

# An integrated health management model to improve the health of professional e-sports athletes: a literature review

Yunxuan Mi[1], Siyuan Zhao[2] and Fangyuan Ju[1]

[1] College of Physical Education, Yangzhou University, Yangzhou, Jiangsu, China
[2] Faculty of Humanities and Arts, Macau University of Science and Technology, Macau, China

## ABSTRACT

**Background**. With the rapid development of the esports industry, an increasing number of esports athletes face various health issues due to occupational characteristics such as a prolonged sedentary lifestyle, high-intensity training, and multi-cycle competitions. Effectively managing and improving the health status of esports athletes has become an urgent need. As a systematic and multidisciplinary collaborative management strategy, the Integrated Health Management Model has been widely applied to various occupational groups, but its application among esports athletes has not yet been systematically reviewed.

**Methodology**. This study adopted a narrative literature review method to collect and analyze the existing literature on the health issues and health management of esports athletes. This study aimed to identify the major health problems of esports athletes and explore the role of the Integrated Health Management Model in improving and preventing these health problems.

**Results**. This study resulted in two main findings: (1) The main health issues faced by esports athletes include musculoskeletal disorders, visual fatigue, metabolic disorders, and psychological stress; (2) the Integrated Health Management Model, through the integration of physical activity and fitness training, psychological counseling, ergonomic optimization, vision protection, and health education, can effectively alleviate common health problems among esports athletes and promote overall health improvement.

**Conclusion**. The Integrated Health Management Model can effectively improve the overall health level of esports athletes through the integration and synergy of multiple strategies.

Corresponding author
Fangyuan Ju, 007882@yzu.edu.cn

# INTRODUCTION

Esports has developed rapidly in recent years to become a large, global sports and entertainment industry. Esports refers to competitive activities conducted through electronic devices, involving teams or individual competitions, especially those among professional gamers (*Giakoni-Ramírez, Merellano-Navarro & Duclos-Bastías, 2022*). With the rapid expansion of digital technology and network communication, esports has gained global attention, building a large user base and audience and becoming an important

platform for professional players, event organizers, and sponsors. In China, the esports industry has also undergone a bumpy development process, but now holds a significant share of the global esports market. In the first half of 2023, the revenue of China's esports industry reached 75.993 billion yuan, with an esports user base of 487 million. Esports have not only injected new vitality into entertainment economic development but also changed people's perception of traditional sports (*China Internet Network Information Center, 2023*).

However, alongside the booming esports industry, the health issues of professional esports athletes have drawn increased attention. Professional e-sports athletes need to stay focused for long periods of time during training and competition, spending an average of 5.5 to 10 h a day playing games, they spend a large amount of time each day performing repetitive postural movements, and their work style is usually characterized by sedentary behavior, long periods of time staring at the screen, and frequent use of hand muscles (*Emara et al., 2020*; *Schary, Jenny & Koshy, 2022*). This long-term, high-intensity work can lead to various health problems. The physical burden and psychological stress of professional esports athletes can affect their professional performance, which may shorten their careers, thereby impacting the sustainable development of the esports industry (*Sanz-Matesanz, Gea-García & Martínez-Aranda, 2023*). Most esports clubs do not have the same health management approaches as traditional sports teams, and the injuries sustained by esports athletes differ significantly from those of traditional athletes, making traditional health management methods less applicable (*DiFrancisco-Donoghue et al., 2019*; *Voisin, Besombes & Laffage-Cosnier, 2022*). Therefore, identifying how to effectively manage the health issues of professional esports athletes has become a major challenge in the esports industry.

The Integrated Health Management Model is based on multidisciplinary collaboration and holistic health management and is widely applied in various fields, with the goal of improving the overall health of individuals. Its core features include multidisciplinary collaboration, patient-centered health services, and continuous and coordinated health management (*DiFrancisco-Donoghue et al., 2019*). This model integrates strategies such as physical exercise, psychological counseling, dietary management, and environmental optimization to improve health through a combination of preventive health management and interventional treatments. The application of the Integrated Health Management Model typically involves close cooperation among various professionals, such as healthcare experts, psychologists, physiotherapists, nutritionists, and social workers (*Ouwens et al., 2005*). This model emphasizes not only the improvement of physical health, but also the comprehensive management of psychological health, lifestyle, and social support, thereby forming a holistic health management system. Under this model, health management professionals can share patient health information, coordinate treatment plans, and continuously track changes to patient health, thereby improving the overall health and quality of life of the patient (*Mäenpää et al., 2009*).

Although the Integrated Health Management Model has achieved positive health outcomes in other occupational groups, its application in the esports field still lacks in-depth research. This study aims to identify the major health issues of professional

**Table 1  PICOS model.** The left column shows the included elements and the right column shows the specific description of each element.

| Element | Description |
| --- | --- |
| Participants | Professional (including youth training) esports athletes |
| Interventions | Studies exploring the impact of the Integrated Health Management Model on health |
| Comparisons | Includes baseline measurements or comparisons with non-intervention groups |
| Outcomes | Studies reporting health impacts, with direct discussion of health dimensions |
| Study Design | Includes randomized controlled trials and observational studies |

esports athletes and explore the role of the Integrated Health Management Model in improving and preventing health problems in this population.

## SURVEY METHODOLOGY

### Search strategy

This study adopted a narrative literature review method, with literature searches conducted through databases such as CNKI, PubMed, SPORTDiscus, PsycINFO, and Web of Science. The search cutoff date was set to September 25, 2024. The search keywords included: "Integrated Health Management Model", "Professional Esports", "Athlete Health", "Physical Activity", "Mental Health", and "Exercise Intervention", among others. During the search process, keyword combinations were used, and Boolean operators (such as "AND", "OR", "NOT") were applied to optimize the search terms, ensuring the results were both comprehensive and accurate. The specific search strategy is outlined, as follows:

(1) **Keyword combinations:**

"Integrated Health Management Model" was combined with "Professional Esports" using the Boolean operator "AND", such as in the search string: "Integrated Health Management Model AND Professional Esports". The "OR" operator was used to connect related concepts, such as "Athlete Health OR Esports Athlete Health".

(2) **Exclusion search settings:**

The "NOT" operator was used to exclude studies unrelated to the topic, such as "Professional Players NOT Amateur Players", to ensure that the search results focused on professional or semi-professional esports athletes.

In addition, reference lists, citation searches, and manual searches were conducted to identify potentially overlooked relevant studies.

### Inclusion and exclusion criteria

This study screened the literature based on the Participants, Interventions, Comparisons, Outcomes, and Study Design (PICOS) model. The specific criteria are shown in Table 1:

The exclusion criteria were: (1) studies involving amateur esports athletes or non-esports populations; (2) literature that had not undergone peer review; (3) studies where the full text was unavailable.
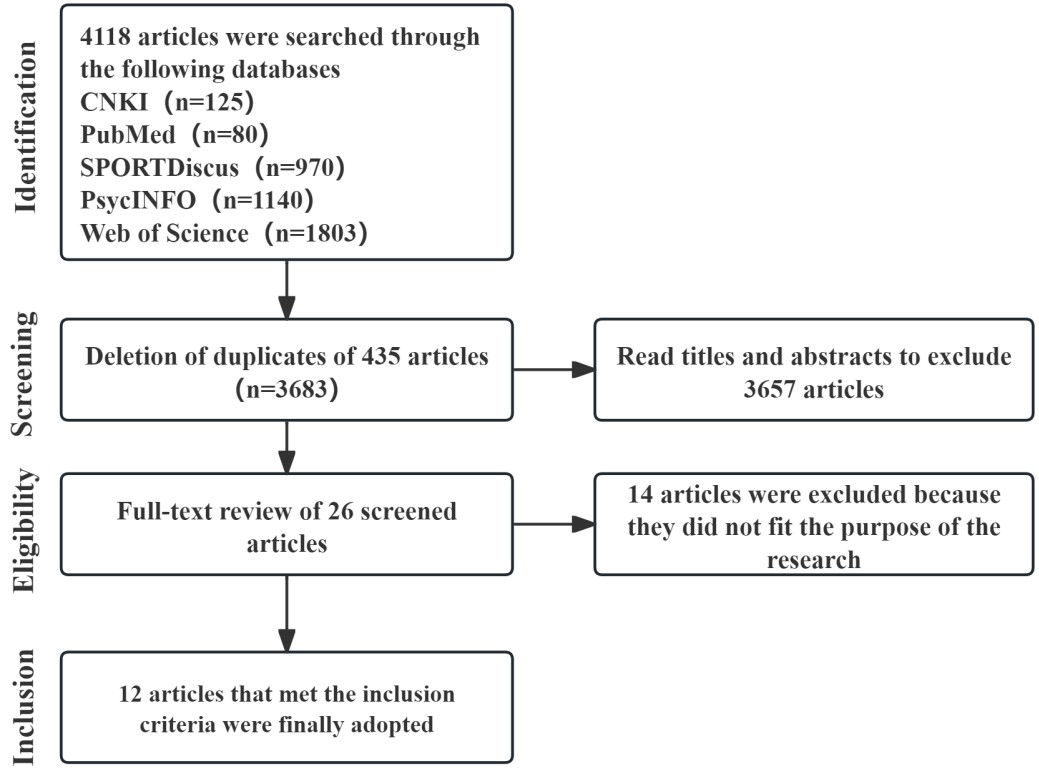

**Figure 1** Narrative literature review article screening process.

## Literature screening process

The literature screening process is shown in Fig. 1. The initial search yielded a total of 4,675 articles. After removing duplicate articles, 3,683 articles remained. The titles and abstracts of these articles were then screened for relevance, resulting in the exclusion of 3,657 unrelated articles. The remaining 26 articles were then subjected to full-text evaluation, where eight articles were excluded because they did not mention the relationship between physical activity/exercise and athlete health, and six articles were excluded because the study designs did not meet the criteria. The remaining 12 articles that met the criteria were included in the analysis.

## Data extraction and analysis

Basic information was extracted from the included studies, including participants, interventions, and main outcomes. The extracted information was recorded using a standard data table for summary and analysis. This study primarily focused on evidence of the impact of physical activity and exercise participation on the health of professional esports athletes, with a comprehensive discussion combining qualitative and quantitative analyses. Table 2 presents a summary of the relevant information from the 12 selected studies.

**Table 2 Summary of literature on health and health management of professional esports athletes.** The table contains the title of these documents, authors, year, what was studied, who was studied, methodology/content of the study, main findings of the study

| Title | Author(s) | Year | Classification of research topics | object of research | Research methodology/content | Main findings |
|---|---|---|---|---|---|---|
| Comparing health risks andmusculoskeletal issues betweenprofessional and casual mobileesports players: a cross-sectionaldescriptive study in Jakarta | Kurniawan et al. | 2024 | Health Status of Professional Esports Athletes | Mobile esports athletes (professionals and recreational) | Cross-sectional descriptive study evaluating fitness, health risks, musculoskeletal and eye issues. | Both professional and recreational athletes face musculoskeletal and visual issues. Professionals show higher physical activity and flexibility. |
| "Constant Pressure of Having to Perform": Exploring Player Health Concerns in Esports | Madden & Harteveld | 2021 | Health Status of Professional Esports Athletes | Professional esports players | Mixed-method study (semi-structured interviews and online surveys) | Esports athletes primarily face cognitive, sleep, and mental health issues due to prolonged screen use. Mindfulness, ergonomics, and socio-emotional learning can help mitigate these issues. |
| Physical symptoms among professional gamers within eSports, a survey study | Ekefjard, Piussi & Hamrin Senorski | 2024 | Health Status of Professional Esports Athletes | Professional esports players | Quantitative study, using questionnaires to assess physical symptoms and lifestyle factors of esports athletes | 62.5% of esports athletes experienced at least one physical symptom in the past three months, including headache and eye discomfort. Gaming over 35 h significantly increases the incidence of physical symptoms. |
| Perspectives of eFootball Players and Staf Members Regarding the Efects of Esports on Health: A Qualitative Study | Monteiro et al. | 2023 | Health Status of Professional Esports Athletes | E-football athletes and staff | Qualitative study (semi-structured interviews), exploring views of athletes and staff on health impacts and improvement strategies | Participants believe esports impacts both mental and physical health, suggesting training optimization, lifestyle improvement, and facility enhancement. |
| Perspectives of Elite Esports Players and Staff Members Regarding the Effects of Esports on Health–a Qualitative Study | Pereira et al. | 2023 | Health Status of Professional Esports Athletes/ Management Model | Elite esports athletes and staff | Qualitative study (semi-structured interviews), exploring athletes' and staff's views on health impacts and improvement strategies | Elite esports athletes and staff identified esports as impacting mental and physical health, recommending training optimization, lifestyle changes, social support, and facility improvements. |
| Health Risks and Musculoskeletal Problems of Elite Mobile Esports Players: a Cross-Sectional Descriptive Study | Lam et al. | 2022 | Health Status of Professional Esports Athletes | Elite mobile esports athletes | Cross-sectional descriptive study investigating health status, fatigue, pain, and musculoskeletal problems | Over 90% of athletes reported fatigue and eye discomfort; over 30% reported headaches and rhinitis, emphasizing the health risks for esports athletes. |
| Examining the Predictors of Mental Ill Health in Esport Competitors | Smith et al. | 2022 | Health Status of Professional Esports Athletes/ Management Model | Collegiate esports athletes | Online questionnaire investigating stressors, sleep quality, burnout, and social anxiety among esports athletes | Stress, sleep quality, burnout, and social anxiety significantly predicted mental health issues in esports athletes, suggesting targeted interventions to improve mental health. |
| Integrated Analysis of Health Dynamics in eSports: Injury ProfilesIntervention Strategies, and Health Optimization Protocols | Mondal & Nithish | 2024 | Health Status of Professional Esports Athletes/ Management Model | Professional esports players | Literature review analyzing health dynamics of esports athletes, including injury profiles, intervention strategies, and health optimization plans | Common issues among esports athletes include musculoskeletal problems, mental health issues, and visual fatigue, with comprehensive intervention strategies proposed. |
| Effect of Ergonomic Training and Exercise in Esports Players: A Randomized Controlled Trial | Gürgan & Şevgin | 2024 | Role of Health Management Model | Professional and recreational esports athletes | Randomized controlled trial evaluating the effects of ergonomic training and exercise on neck and upper limb function, and sleep quality among esports athletes | Ergonomics-based exercise significantly improves neck and upper limb function, as well as sleep quality among esports athletes. |

**Table 2** (*continued*)

| Title | Author(s) | Year | Classification of research topics | object of research | Research methodology/content | Main findings |
|---|---|---|---|---|---|---|
| Professional Esports Players: Motivation and Physical Activity Levels | *Giakoni-Ramírez, Merellano-Navarro & Duclos-Bastías* | *2022* | Health Status of Professional Esports Athletes | Professional esports athletes | Cross-sectional and observational study investigating motivation and physical activity levels of professional esports athletes in Europe and Latin America | 92.7% of professional esports athletes reported moderate to high levels of physical activity, with intrinsic and extrinsic motivation negatively correlated with energy expenditure. |
| eSports: the need for a structured support system for players | *Hong* | *2023* | Role of Health Management Model | Esports athletes and stakeholders | Semi-structured interviews analyzing the views of 51 participants on esports athletes' health and the need for support systems | Emphasizes the need for esports athletes to balance training and education, proposing a holistic development model to support their professional and health development. |
| Managing the health of the eSport athlete: an integrated health management model | *DiFrancisco-Donoghue et al.* | *2019* | Role of Health Management Model | Esports athletes | Electronic questionnaire evaluating 65 esports athletes in the U.S. and Canada, proposing an integrated health management model | Esports athletes' primary health issues include eye fatigue, neck and back pain, and hand pain, with 40% not participating in any form of physical activity. An integrated health management approach is recommended. |

## Quality assessment

To ensure reliability, the "Standard Quality Assessment Criteria" (*Cook, Kmet & Lee, 2004*) was used to evaluate the quality of the included studies, focusing on the accuracy of the study design, statistical analysis, and results reporting. All assessments were independently conducted by two researchers, and any discrepancies were resolved through discussion.

## RESULTS

### Health status of professional esports athletes

As an emerging professional group, the health status of professional esports athletes has increasingly drawn attention from both academia and the esports industry. Similar to traditional professional athletes, professional esports athletes face a high risk of physical injuries, and the occupational characteristics of esports increase the risk of additional health issues. Professional esports athletes are subjected to high-intensity training and competition environments, requiring prolonged sedentary behavior and fixed postures, which expose these athletes to multiple musculoskeletal, visual, metabolic, and mental health risks.

### Musculoskeletal injuries among professional esports athletes due to high-intensity training

Prolonged competitions and training sessions require professional esports athletes to maintain a fixed sitting posture while using upper limbs, neck, shoulder, and lower back muscles. The average weekly training time of esports athletes often exceeds 35 h, significantly increasing the incidence of musculoskeletal issues, making musculoskeletal problems one of the most common health issues among esports athletes (*Ketelhut et al., 2023*; *Tang et al., 2023*). Studies have shown that over 85% of professional esports athletes report experiencing varying degrees of musculoskeletal pain and discomfort during competitions

and training, with the most pronounced pain occurring in the neck, shoulders, lower back, and wrists (*Lam et al., 2022*).

During competitions, esports athletes need to maintain a fixed posture and intense focus for long periods of time, resulting in tension in the neck and shoulder muscles, which can cause muscle fatigue, chronic pain, and stiffness (*Kurniawan et al., 2024*). Some esports athletes report experiencing neck stiffness and shoulder soreness after prolonged competitions, with such pain often worsening as training duration increases (*Madden & Harteveld, 2021*; *Monteiro et al., 2023*).

Chronic lower back strain is also a common health issue among professional esports athletes. During competitions and training, these athletes typically maintain a sedentary posture, with limited movement of the lower back muscles, leading to prolonged muscle tension, chronic strain, and lumbar disc herniation (*Ekefjard, Piussi & Hamrin Senorski, 2024*). Studies have shown that over 60% of professional esports athletes experience lower back pain and stiffness after prolonged competitions (*Monteiro et al., 2023*). These issues are common among esports athletes and tend to be severe; without effective recovery methods and treatment measures, these injuries could have long-term impacts on both the careers and quality of life of esports athletes (*Kurniawan et al., 2024*).

Carpal tunnel syndrome is another common musculoskeletal issue among professional esports athletes. The frequent use of keyboards and mice during competitions and training makes the repetitive movements of the wrists highly susceptible to carpal tunnel syndrome and tendonitis (*Wattanapisit, Wattanapisit & Wongsiri, 2020*; *Yin et al., 2020*). Typical symptoms of carpal tunnel syndrome include finger numbness, wrist pain, and decreased hand strength (*Kurniawan et al., 2024*; *Wattanapisit, Wattanapisit & Wongsiri, 2020*). In a survey study on professional esports athletes, over 50% reported experiencing finger numbness and wrist soreness during competitions and training (*Ekefjard, Piussi & Hamrin Senorski, 2024*). If not promptly addressed and treated, these symptoms may result in impaired finger function among professional esports athletes and could affect their operational abilities and competitive performance (*Kurniawan et al., 2024*; *Lam et al., 2022*). These symptoms may initially manifest as mild muscle soreness, but without timely intervention and treatment, they could progress into chronic inflammation, tendon tears, and joint degeneration (*Shen & Cicchella, 2023*).

The nature of esports requires athletes to maintain intense focus during competitions, often leading them to overlook physical fatigue and injuries, which may ultimately exacerbate musculoskeletal damage (*Ketelhut et al., 2023*). Therefore, musculoskeletal problems are among the most prevalent health issues for professional esports athletes, affecting both their performance and career development.

## Visual fatigue and vision problems as common health concerns among professional esports athletes

Professional esports athletes must stare at electronic screens for extended periods, making visual fatigue and vision problems among the most common health concerns in this group. During competitions and training, esports athletes usually maintain prolonged, intense focus, leading to sustained tension in the eye muscles, resulting in symptoms

such as dry eyes, blurred vision, and eye soreness (*Ekefjard, Piussi & Hamrin Senorski, 2024*; *Wattanapisit, Wattanapisit & Wongsiri, 2020*). Studies indicate that over 90% of professional esports athletes have experienced eye discomfort after competitions or training (*Monteiro et al., 2023*; *Yin et al., 2020*). These symptoms impact short-term visual health, and prolonged exposure to the high-energy blue light that electronic devices emit may damage the photoreceptor cells and retina, leading to more serious vision problems (*Lam et al., 2022*; *Madden & Harteveld, 2021*).

The incidence of visual fatigue among professional esports athletes is significantly higher than that of the general population, primarily due to prolonged exposure to blue light radiation and high-intensity visual concentration in front of screens (*Ekefjard, Piussi & Hamrin Senorski, 2024*; *Shen & Cicchella, 2023*; *Yin et al., 2020*). The blue light from these screens increases the risk of macular degeneration and vision decline (*Lam et al., 2022*; *Monteiro et al., 2023*). A study on the visual health of professional esports athletes showed that prolonged exposure to blue light radiation may cause retinal cell damage and irreversible vision loss (*Ekefjard, Piussi & Hamrin Senorski, 2024*). In another survey of professional esports athletes, over 70% reported experiencing dry eyes and temporary vision decline, primarily due to prolonged exposure to strong screen light and a significantly reduced blink rate (*Pereira et al., 2023*). When the eyes experience fatigue from prolonged high-intensity focus, their accommodation ability gradually decreases, making it difficult to quickly adjust focus (*Ekefjard, Piussi & Hamrin Senorski, 2024*; *Wattanapisit, Wattanapisit & Wongsiri, 2020*). Some esports athletes report experiencing blurred vision, eye soreness, and temporary loss of focus after training and competitions, which not only affects their short-term competitive performance but may also cause irreversible damage to their long-term visual health (*Madden & Harteveld, 2021*; *Yin et al., 2020*).

Prolonged visual fatigue manifests as eye discomfort and pain and can result in decreased accommodation ability among professional esports athletes (*Monteiro et al., 2023*; *Yin et al., 2020*). Research indicates that during high-intensity competitive seasons, the accommodation ability and focus-switching speed of professional esports athletes are significantly lower than those of the general population, with some athletes experiencing temporary blurred vision and increased intraocular pressure (*Ekefjard, Piussi & Hamrin Senorski, 2024*; *Lam et al., 2022*). Some athletes have also reported experiencing blurred vision, double vision, and chronic eye fatigue during prolonged training periods (*Pereira et al., 2023*; *Yin et al., 2020*). Though these symptoms often subside after the competition or training period, if these issues are not effectively managed, they could further develop into serious eye diseases such as glaucoma, cataracts, and retinal degeneration (*Pereira et al., 2023*; *Wattanapisit, Wattanapisit & Wongsiri, 2020*; *Madden & Harteveld, 2021*; *Monteiro et al., 2023*).

To address visual fatigue and vision problems, some professional esports athletes choose to wear blue light-blocking glasses or regularly use artificial tears to alleviate symptoms. However, these measures only provide short-term relief from eye discomfort and do not fundamentally prevent the occurrence of visual fatigue (*Ekefjard, Piussi & Hamrin Senorski, 2024*; *Lam et al., 2022*). Vision problems among professional esports athletes remain difficult to effectively resolve.

## Metabolic disorders are common and increasingly severe among professional esports athletes

Metabolic disorders are particularly prevalent among professional esports athletes because of their prolonged sitting, lack of regular physical activity, and poor dietary habits. Compared to the general population and other athletes, professional esports athletes have significantly increased risks of obesity, hypertension, hyperglycemia, hyperlipidemia, and other metabolic syndromes (*Monteiro et al., 2023*; *Yin et al., 2020*), it certainly increases their risk of death. Studies have shown that over 40% of professional esports athletes exhibit symptoms of metabolic disorders, with overweight and abnormal fat metabolism being particularly common (*Wattanapisit, Wattanapisit & Wongsiri, 2020*).

The metabolic disorders of professional esports athletes first manifest as insufficient energy expenditure and fat accumulation caused by prolonged sitting (*Madden & Harteveld, 2021*; *Verdú Navarro, Homs & Bodas-Vaello, 2021*). Professional esports athletes have a significantly lower basal metabolic rate compared to the general population (*Wattanapisit, Wattanapisit & Wongsiri, 2020*). Prolonged sitting not only reduces the body's energy expenditure but also causes fat accumulation in the abdomen, forming visceral fat (*Verdú Navarro, Homs & Bodas-Vaello, 2021*). One study found that nearly 30% of professional esports athletes have experienced rapid weight gain and abdominal obesity during their careers (*Madden & Harteveld, 2021*).

The unhealthy eating habits of esports athletes during competitions and training can exacerbate the severity of metabolic disorders (*Martinez et al., 2021*). To maintain energy during long competitions, esports athletes often choose high-sugar, high-fat foods and beverages, such as energy drinks, sodas, and sugary snacks (*Madden & Harteveld, 2021*). These foods are high in calories and cause a rapid increase in both blood sugar levels and insulin resistance, which are both closely related to the occurrence of metabolic syndrome (*Basciano, Federico & Adeli, 2005*). Research shows that over 50% of professional esports athletes have experienced blood sugar fluctuations and abnormal blood lipid levels due to high-sugar diets during their careers (*Ekefjard, Piussi & Hamrin Senorski, 2024*; *Pereira et al., 2023*). In addition, high-sugar diets can lead to fat accumulation on the walls of blood vessels, increasing the risk of cardiovascular diseases (*Heber, 2010*).

Irregular eating and sleeping patterns further aggravate metabolic problems among esports athletes (*King et al., 2020*). The competition schedules of professional esports athletes are typically concentrated during prime viewing hours in the evening and nighttime. In the absence of matches, training times range from afternoon to early morning, with some athletes training until four or five o'clock in the morning (*Pluss et al., 2022*). During late-night training, athletes must eat to maintain their basic competitive state (*Bonci, 2011*). The reversal of day and night, lack of sleep, and irregular eating patterns all disrupt hormone secretion and metabolic balance in the body, eventually leading to weight gain and worsening metabolic disorder symptoms (*Kim, Jeong & Hong, 2015*).

Metabolic disorders among professional esports athletes are the result of multiple factors, such as sedentary behavior, high-sugar diets, irregular routines, and lack of physical activity. The presence of these metabolic problems affects the health status of esports athletes and may lead to serious chronic diseases.

## Mental health issues are widespread and difficult to manage among professional esports athletes

Professional esports athletes often face immense psychological pressure during high-intensity training and competitions, making mental health issues widespread and difficult to effectively manage among esports athletes. These issues manifest as anxiety, depression, mood swings, sleep disorders, occupational burnout, and various other mental health problems (*KuMari, Sharma & Singh, 2022*). Compared to other professional athletes, esports athletes have a higher incidence of mental health issues and are also more susceptible to the stresses of external environmental factors and career uncertainties (*Palanichamy et al., 2020*).

Due to the high levels of competitiveness and career uncertainty in esports, professional esports athletes are prone to anxiety and depression (*Palanichamy et al., 2020*), During competitions, esports athletes face the high expectations of coaches, teammates, and spectators while dealing with rapidly changing competitive environments, keeping these athletes in a prolonged state of mental tension. Research indicates that over 60% of professional esports athletes exhibit symptoms of anxiety and depression during the competitive season (*Sousa et al., 2020*; *Wattanapisit, Wattanapisit & Wongsiri, 2020*). This anxiety can affect the short-term competitive performance of these athletes and may lead to serious mental disorders in the long term (*Lam et al., 2022*). The career span of esports athletes is relatively shorter than the careers of professional athletes of other sports, with most esports athletes facing career bottlenecks and retirement pressure early or mid-career, further exacerbating their anxiety and depression (*Bányai et al., 2020*; *Meng-Lewis et al., 2022*). Some esports athletes report feeling helpless and confused due to the instability of their performance and uncertainty about their future development (*Butt & Molnar, 2009*).

Public opinion and social media assessments pose significant mental health challenges for professional esports athletes (*Leis et al., 2022*). Professional esports athletes face criticism and negative comments from both their audiences and the media when they fail in competitions or their performance declines (*Hou, Yang & Panek, 2020*; *Ruvalcaba et al., 2018*). Studies show that some professional esports athletes experience online abuse, including malicious comments on social media, leading to increased emotional fluctuations and decreased self-esteem (*Monteiro et al., 2023*). This external psychological pressure intensifies the anxiety and depression of these athletes and may generate more negative emotions during competitions, affecting the overall performance of esports athletes. Some esports athletes report feeling increased psychological pressure due to negative comments after a loss, which may cause these athletes to lose confidence and enthusiasm for competitions (*Neil et al., 2011*; *Pereira et al., 2023*). This excessive, prolonged psychological burden can also lead to occupational burnout among esports athletes (*Carrani et al., 2022*; *Ekefjard, Piussi & Hamrin Senorski, 2024*). Occupational burnout is typically characterized by emotional detachment, loss of enthusiasm for competitions, and decreased work motivation. Once occupational burnout sets in, it is often difficult to alleviate in a short period of time, potentially leading to a decline in athletic performance and an increase in thoughts of retirement (*Potter, 2009*; *Schaufeli, Maslach & Marek, 2017*).

In summary, the mental health issues of professional esports athletes result from both occupational characteristics and the combined effects of social opinion and professional pressure. These mental health issues affect the competitive performance of these athletes and may have serious negative impacts on their careers and personal lives. However, due to the concealed nature of mental health issues and the lack of effective management measures, most esports athletes tend to silently endure these struggles or rely on personal willpower to cope, rather than seeking professional psychological assistance. Therefore, the mental health issues of professional esports athletes urgently require more systematic attention and management.

## The role of the integrated health management model in promoting the health of professional esports athletes

The occupational health issues of professional esports athletes are distinct, mainly manifesting as musculoskeletal injuries, visual fatigue, metabolic disorders, and mental health problems. The interaction and cumulative effects of these health problems severely impact the competitive performance of professional esports athletes and may lead to shortened careers and poor overall health. Therefore, effectively managing and improving these health issues has become a primary focus for current esports clubs.

Traditional single-method health management approaches struggle to address the complex and multidimensional nature of the health issues professional esports athletes face. Therefore, the Integrated Health Management Model, as a systematic and multidisciplinary collaborative health management strategy, can play a more significant role in improving the health of professional esports athletes (*DiFrancisco-Donoghue et al., 2019*). By integrating interventions such as physical exercise, psychological counseling, nutritional management, and work environment optimization, this model can help effectively alleviate musculoskeletal issues, visual fatigue, and metabolic disorders among professional esports athletes and help them cope with mental health challenges in high-pressure environments, thereby enhancing their overall health and career development potential.

Based on the occupational and behavioral characteristics of professional esports athletes, as well as their exhibited health issues, an application of the Integrated Health Management Model in esports should include the following core components: physical activity and fitness training, mental health management, ergonomic optimization, vision protection, and health education and lifestyle management (*DiFrancisco-Donoghue et al., 2019*).

## Physical activity and fitness training improves physical fitness and reduces injury risk

Physical activity and fitness training are essential components of the Integrated Health Management Model, playing a crucial role in improving the overall strength of professional esports athletes and reducing their risk of musculoskeletal injuries. The core concept of the Integrated Health Management Model is to improve the health of professional esports athletes through systematic, multidimensional health management strategies, thereby enhancing their professional competitiveness and overall health. In this model, physical activity and fitness training serve both as standalone interventions to alleviate and prevent health issues and as combined interventions with other health management measures

to collectively enhance the overall health of professional esports athletes (*DiFrancisco-Donoghue et al., 2019*).

Physical activity and fitness training serve as the "foundation for health assurance" in the Integrated Health Management Model, forming the cornerstone that supports the effective implementation of other health management measures. Good physical fitness is central to the health management of professional esports athletes, providing a solid physical foundation for the management of their daily health (*Haskell, Montoye & Orenstein, 1985*). Through fitness training, professional esports athletes can improve their overall physical function and increase their muscle strength, endurance, and cardiovascular capacity, enhancing their ability to meet high-intensity professional demands and effectively reducing the negative impact of physical fatigue or health issues on their mental state and emotional regulation (*Dykstra, Koutakis & Hanson, 2021*; *McNulty et al., 2023*). After high-intensity competitions, professional esports athletes often experience physical fatigue and muscle stiffness, which affects their professional performance and exacerbates emotional fluctuations and occupational burnout (*Akyüz, 2022*). Therefore, physical activity and fitness training provide the foundational support for esports athletes to maintain a stable mental state and high-level performance in high-pressure environments (*McNulty et al., 2023*). Additionally, physical activity and fitness training play a proactive intervention and prevention role within the Integrated Health Management Model. Fitness training programs can prevent common musculoskeletal injuries such as lower back pain, neck and shoulder strain, and carpal tunnel syndrome among professional esports athletes (*Kocić et al., 2022*; *Lam et al., 2022*). Aerobic exercise within fitness training programs also promotes cardiovascular health and metabolic function, reducing obesity and other metabolic diseases caused by prolonged sitting and irregular routines (*Chapman et al., 2013*). This proactive intervention strategy helps alleviate the existing health issues of esports athletes and enhances their muscle endurance and flexibility, reducing their risk of retirement or performance decline due to health problems (*Sanz-Matesanz, Martínez-Aranda & Gea-García, 2024*). As foundational and indispensable components of the Integrated Health Management Model, physical activity and fitness training play an important role in reducing musculoskeletal injuries and improving the overall physical fitness of professional esports athletes.

## Mental health management reduces anxiety and depression and enhances stress resilience

Mental health management is a key component of the Integrated Health Management Model. The core role of mental health management is to enhance the psychological resilience of professional esports athletes through systematic psychological interventions and emotion management strategies. The goal of mental health management in esports athletes is to alleviate the anxiety, depression, and burnout caused by occupational stress and competitive environments, thereby improving the stress resilience and emotional regulation of these athletes throughout their careers (*Purcell, Gwyther & Rice, 2019*; *Smith et al., 2022*; *Soares, Goedert & Vargas, 2022*).

When professional esports athletes face competitive losses, poor performance, or career bottlenecks, they often experience high levels of anxiety and depression, which affect their professional performance and may lead to serious mental health issues. Through Cognitive Behavioral Therapy (CBT) and psychological counseling, professional esports athletes can learn to identify the sources of their emotions and adopt more positive ways to deal with negative emotions, thereby reducing the negative impact of anxiety and depression (*Ellis & Korman, 2022*; *Rossoni et al., 2023*). CBT also helps professional esports athletes develop positive coping strategies, enhancing their emotional regulation and enabling them to maintain a better mental state in high-pressure environments (*Podlog et al., 2020*). Additionally, mental health management strategies emphasize building group support and social networks, helping professional esports athletes establish a stable social support system that can provide emotional support and psychological comfort during mental crises (*Ellis & Korman, 2022*).

Mental health management strategies also play an important role in enhancing the stress resilience of professional esports athletes. Professional esports athletes, who are often in high-pressure environments, frequently experience emotional outbursts due to excessive stress (*Reardon et al., 2021*). Mental health management strategies, including stress management courses, emotion regulation training, and crisis intervention techniques, help professional esports athletes use scientific relaxation exercises and breathing techniques to transform stress into positive motivation (*Smith et al., 2022*). Participating in stress management courses can significantly reduce the emotional fluctuations of esports athletes and improve their psychological recovery, aiding in rapid emotional recovery and mental rebuilding after long, intense competitions. Regular relaxation training can lower cortisol levels in professional esports athletes, reducing physiological stress responses caused by pre-competition anxiety, enabling the athletes to maintain both calmness and focus during high-intensity competitions (*Cruess et al., 2000*; *Morrey, 1996*; *Reardon et al., 2021*).

Mental health management can also prevent occupational burnout among professional esports athletes. Occupational burnout is a common mental health issue for esports athletes, mainly characterized by emotional exhaustion, reduced sense of achievement, and loss of interest in competitions and training (*Smith et al., 2022*). Regular psychological assessments, counseling, and personalized recovery training, along with the use of the Burnout Assessment Tool, enable comprehensive evaluation of the mental health status of professional esports athletes. This approach involves identifying early signs of burnout and then developing personalized intervention plans based on assessment results, effectively preventing burnout and aiding in timely recovery during mental crises (*Markati et al., 2019*; *Reardon et al., 2021*; *Schaufeli, Desart & De Witte, 2020*). Moreover, the mental health management strategies of meditation and mindfulness practices can enhance the emotional regulation of esports athletes, helping them cope more calmly with career bottlenecks and reducing their risk of occupational burnout (*Rogowska & Tataruch, 2024*).

As a core component of the Integrated Health Management Model, mental health management provides comprehensive and effective psychological support for professional esports athletes through various strategies such as emotional regulation, stress management, and burnout prevention. These methods help reduce anxiety and depression among

professional esports athletes and enhance their stress resilience and psychological toughness, promoting long-term health despite high-pressure environments.

## Ergonomic optimization improves training environment and equipment for professional esports athletes

Ergonomic optimization is an often overlooked part of the Integrated Health Management Model, as it is not typically considered a health measure. The core purpose of ergonomic optimization is to use scientifically designed equipment and environmental layouts to reduce the risk of musculoskeletal issues, nerve compression, and other occupational diseases caused by prolonged poor posture and repetitive movements (*Gürgan & Şevgin, 2024*; *Martorana et al., 2022*).

Using ergonomically designed chairs and monitors can effectively reduce the neck and shoulder muscle strain of esports athletes, while adjusting the position of keyboards and mice can lessen tension in their wrists and forearms. Improving training space layouts, such as raising desk height, adding supportive equipment, and using wrist rests, can effectively reduce the burden on the wrists and shoulders of these athletes, lowering the incidence of carpal tunnel syndrome and cervical spondylosis caused by repetitive actions (*Gürgan & Şevgin, 2024*; *Martorana et al., 2022*; *Tapanya et al., 2021*).

Ergonomic optimization strategies also include posture correction training programs, further enhancing the physical health and work comfort of professional esports athletes. Posture correction training, such as regular core muscle exercises and neck and shoulder stretches, helps professional esports athletes restore normal spinal curvature, alleviate back pain and neck tension caused by forward spinal flexion, prevent lumbar lordosis and thoracic kyphosis due to prolonged poor posture, and maintain proper posture during competitions and training (*Gürgan & Şevgin, 2024*; *Zurawski et al., 2020*). Stretching exercises for the hand and wrist nerves can effectively prevent the occurrence of carpal tunnel syndrome and other hand injuries (*Hong, 2023*).

As a vital component of the Integrated Health Management Model, ergonomic optimization plays a preventive role in maintaining the health of professional esports athletes, enhancing both work comfort and overall health by improving training environments and body posture, thereby reducing occupational health risks.

## Vision protection measures help prevent visual fatigue and decline in professional esports athletes

Professional esports athletes, due to prolonged, high-intensity screen viewing, are highly susceptible to visual fatigue, dry eye syndrome, and other eye problems. Vision protection measures play a crucial preventive role in the Integrated Health Management Model, with core strategies including proper screen brightness and contrast adjustment, blue light-blocking glasses, appropriate screen-viewing distance, and regular eye breaks. Blue light-blocking glasses, which effectively filter out some harmful blue light, combined with suitable screen brightness and distance, can greatly reduce retinal strain from electronic screens, lowering the incidence of blurred vision and eye fatigue (*Al-Mohtaseb et al., 2021*; *Dyrek et al., 2024*). Esports athletes can alleviate eye fatigue by following the "20-20-20" rule: every 20 min, look at something at least 20 ft away for 20 s. Regularly using eye drops

to keep the eyes moist can help prevent dry eye syndrome (*Ellis & Korman, 2022*). Over time, these practices can help professional esports athletes increase their awareness of eye health and develop good eye habits, preventing vision decline and other long-term eye problems. These healthy eye habits can also help esports athletes maintain visual comfort for longer periods during competitions, enhancing their competitive performance (*Moe et al., 2023*).

Therefore, in the integrated health management of professional esports athletes, vision protection should be regarded as equally important as physical training and mental health management and should receive adequate attention and application in daily training and competition management.

## Health education and lifestyle management promote health awareness and the development of good habits

Health education on topics such as diet, sleep management, personal hygiene, and emotional and stress management, can enhance athletes' awareness of personal health and guide them to develop healthier behavior patterns and lifestyle habits throughout their careers. Health education aims to help athletes build positive health beliefs and behaviors and reduce the frequency of unhealthy behaviors, thereby lowering potential health risks (*Nutbeam, 2000*).

Health education can help professional esports athletes recognize the negative effects sedentary behavior and lack of physical activity have on their health. Health education can further encourage these athletes to enhance their physical fitness through daily, moderate exercise and fitness training, thereby reducing their risk of obesity, metabolic disorders, and musculoskeletal injuries (*Wood & Neal, 2016*). Nutritional management courses can guide athletes on how to choose a healthy diet, avoid excessive intake of high-sugar and high-fat foods, and prevent metabolic diseases (*ALamari, 2020*). Lifestyle management introduces health interventions such as scheduling adequate rest periods, improving sleep quality, and using scientific recovery methods to improve the overall health of professional esports athletes (*Calleja-González et al., 2016*).

Systematic health education and lifestyle management can fundamentally change the health awareness and behavior patterns of professional esports athletes, providing them with long-term health protection. Continuous health education can improve these athletes' ability to monitor their own health status (*Lima et al., 2018*), further encouraging them to adopt healthier lifestyles, such as reducing sedentary time, increasing physical activity, adjusting diet, and undergoing regular health check-ups.

However, health education and lifestyle management are relatively difficult to implement in the Integrated Health Management Model for esports, as improving health awareness is a long-term process that does not yield immediate results, making it challenging for many athletes and clubs to maintain commitment.

## The mutual promotion among strategies in the integrated health management model

In managing the health of professional esports athletes, the value of the Integrated Health Management Model lies not only in the independent application of each health intervention

strategy, but also in the compounded effect of these interventions. The effective integration of various interventions can maximize the effectiveness of each strategy, further enhancing the overall health of the athletes.

Physical activity and fitness training, as foundational strategies of the model, improve athletes' physical fitness and provide strong physical support for other health management measures. Good physical fitness boosts athletes' confidence, and regular physical exercise increases dopamine and endorphin secretion in the brain (*Dishman & O'Connor, 2009*), improving emotional stability and psychological endurance, enhancing mental health management. Participation in physical activities also improves the focus and concentration of professional esports athletes, enabling them to better absorb health education and lifestyle management content and more consciously follow health behavior guidelines (*Sondos, Animesh & Kumar, 2023*).

Mental health management enhances emotional stability and psychological resilience, improving the physical activity, health education, and lifestyle management of professional esports athletes. Emotionally stable athletes show greater focus and persistence in fitness training, with higher execution and self-discipline, leading to better training outcomes. Mental health management also increases athletes' acceptance of health education and lifestyle management, improving their participation in various health management programs and in proactively adjusting their routines, diet, and behaviors.

Ergonomic optimization provides a comfortable and safe training environment for professional esports athletes, enhancing their physical activity and mental health management. Good environmental design reduces the risk of sports injuries during fitness training, making physical activity and fitness training more effective. Ergonomic optimization also reduces the physical burden of professional esports athletes during training and competitions, helping them relax more, thereby improving their emotional regulation and stress management. Good environmental design can improve athletes' focus during health education, helping them to better understand the content. A comfortable environment can also promote lifestyle management, leading athletes to adopt healthy behaviors in daily life.

Vision protection measures improve visual health, enhancing the attention and concentration of esports athletes during physical activity, mental health management, and health education. Good visual health helps professional esports athletes maintain better focus and responsiveness during fitness training, enabling them to complete training plans more effectively. Improved visual health reduces emotional fluctuations caused by visual fatigue, allowing professional esports athletes to maintain a more stable emotional state, improving mental health management.

Health education and lifestyle management help professional esports athletes better understand and accept each health management strategy, improving their overall health. Health education helps professional esports athletes recognize the importance of physical and mental health, motivating them to actively participate in fitness training and maintain a positive mindset. When professional esports athletes understand health management strategies through health education, they show greater adherence in vision protection and

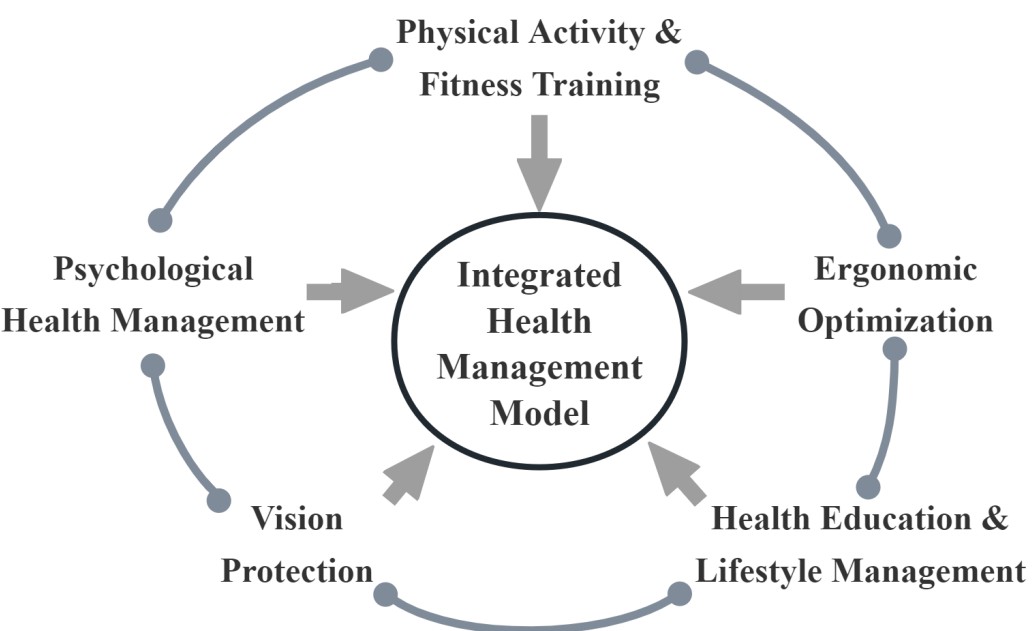

**Figure 2** **Mutual enhancement diagram of health management strategies for professional esports athlete.** The links between the components of the Integrated Health Management Model.

ergonomic optimization and actively engage in various health management measures, enhancing the overall effectiveness of integrated health management.

The strategies in the Integrated Health Management Model mutually reinforce each other (as shown in Fig. 2). The mutual promotion and coordination of individual strategies increase the overall effectiveness of the Integrated Health Management Model. This model can provide comprehensive health protection for professional esports athletes.

# CONCLUSIONS

This study, through a narrative literature review, analyzed the main health issues faced by professional esports athletes and the role of the Integrated Health Management Model in promoting the health of these athletes. The study found that professional esports athletes commonly face musculoskeletal injuries, visual fatigue, metabolic disorders, and mental health issues. The Integrated Health Management Model, through the integration and coordination of multiple strategies, can effectively mitigate these health risks and improve overall health levels. However, empirical research on this model in the esports field remains relatively limited. Future research should integrate big data technology and personalized health management methods to help provide scientific evidence and practical guidance for the health management of professional esports athletes.

# ACKNOWLEDGEMENTS

The authors of this article would like to thank all esports practitioners for their dedication to the industry and all those who follow and support the esports industry.

### Funding
The authors received no funding for this work.

### Competing Interests
The authors declare there are no competing interests.

### Author Contributions
- Yunxuan Mi conceived and designed the experiments, performed the experiments, analyzed the data, prepared figures and/or tables, and approved the final draft.
- Siyuan Zhao conceived and designed the experiments, authored or reviewed drafts of the article, and approved the final draft.
- Fangyuan Ju conceived and designed the experiments, analyzed the data, authored or reviewed drafts of the article, and approved the final draft.

### Data Availability
This is a literature review.

### Supplemental Information
Supplemental information for this article can be found online at http://dx.doi.org/10.7717/peerj.19323#supplemental-information.

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
