# Peer review of "An integrated health management model to improve the health of professional e-sports athletes: a literature review"

_PeerJ, doi:10.7717/peerj.19323_

## Round 0.1 · original submission · Major Revisions

Some revisions are needed before acceptance.

·

Basic reporting

The composition of this paper strikes an appropriate balance between thoroughness and conciseness. It makes for a very clear read and is well-written.

The authors cite relevant and appropriate papers. However, I am surprised that their literature review either did not identify or did not include some relevant research, including the Schary et al., 2022 paper "Leveling Up Esports Health: Current Status and Call to Action" or the Voisin et al., 2022 paper "Are Esports Players Inactive? A Systematic Review".

Experimental design

The study design is appropriate as is their documentation of their methodology.

Validity of the findings

Conclusions are well-stated, well-founded, and appropriately sourced.

Additional comments

I particularly appreciated the comprehensiveness of the Discussion section, which lays out a solid case for the value of multidisciplinary care models in esports.

·

Basic reporting

The review is of broad and cross-disciplinary interest, and is within the scope of the journal.
The field been reviewed recently, but considering different aspects of the problem. I think this review is useful, because awareness on medical and psychological consequences of egames are still under-considered.
The Introduction adequately introduce the subject and make it clear who the audience the motivation of the study.

Experimental design

The employed Survey Methodology is appropriate and consistent with a comprehensive, unbiased coverage of the subject. However, there is no mention in the paper to the high blood pressure caused by extensive hours of playing, neither are cited the deaths associated with egaming. I recommend to expand to cite the papers that reported fatalities in egaming and the effect on cardiovascular system.
Are sources are adequately cited and quoted or paraphrased appropriately.
The review is organized logically into coherent paragraphs/subsections.

Validity of the findings

The paper propose to employ an Integrated Health management approach to egamer’s health. This is an interesting and new approach.
Is there a well developed and supported argument that meets the goals set out in the Introduction? The goals set out in the introduction are the supported appropriately in the text.
The Conclusion pose interesting questions for future research.

Additional comments

There is no mention to the effect of intensive egaming on cardiovascular health, including sudden deaths. I think must be included in this review, or at least mentioned in the introduction.

·

Basic reporting

Title:
• The title is overly long and could be more concise while maintaining clarity.
• There is no mention that this is a literature review, which may mislead readers into expecting original research.

Introduction:
• The introduction section feels superficial and does not sufficiently establish why the proposed Integrated Health Management Model is necessary. Consider expanding on the significance of esports-related health issues with stronger justification. In particular, the following statement lacks numerical support and concrete examples:
"This long-term, high-intensity work can lead to various health problems. The physical burden and psychological stress of professional esports athletes can affect their professional performance, which may shorten their careers, thereby impacting the sustainable development of the esports industry (Sanz-Matesanz et al., 2023)."
• It would be beneficial to include statistics on how common or severe these health issues are. Additionally, citing real-world cases from TSM TheOddOne’s scurvy (2012) to Faker’s wrist injury (2024) would strengthen the argument by demonstrating that even high-level professional players are still affected by these health risks despite advancements in the esports industry.

Experimental design

Methods:
• The methodology is generally well described and formulated, providing a clear overview of the literature search strategy, inclusion criteria, and data extraction process.
• The use of multiple databases (CNKI, PubMed, SPORTDiscus, PsycINFO, Web of Science) strengthens the comprehensiveness of the review.
• The inclusion of Boolean operators and keyword combinations demonstrates a systematic approach to identifying relevant studies.

Validity of the findings

Results:
• A significant portion of the already lengthy Results section consists of theoretical background that would be more appropriately placed in the Introduction. The Results section should primarily focus on summarizing the findings from the reviewed literature, while broader contextual information about esports health risks and management strategies should be introduced earlier. Relocating this content to the Introduction would improve the clarity and structure of the manuscript. Additionally, theoretical concepts can be revisited later, when they can be critically analyzed in relation to the presented results.
• Despite the length of the Results section, the subsections ealth Status of Professional Esports Athletes, Mental health management, Ergonomic optimization, Vision protection measures, and Health education and lifestyle management do not cite a single scientific source. This lack of references undermines the credibility and academic rigor of the content, as key claims are presented without empirical support
Conclusion:
• The Conclusion is clear and well stated, effectively summarizing the key findings of the review. However, the terms "musculoskeletal injuries, visual fatigue, metabolic disorders, and mental health issues" encompass a broad spectrum of conditions. Providing more specific details on the most prevalent or severe issues within these categories would enhance clarity and strengthen the impact of the conclusions. A more precise breakdown would also help readers better understand the specific health risks esports athletes face and how the proposed management model addresses them.

Additional comments

General Comments:
• I congratulate the authors on presenting an interesting and valuable analysis of esports health management. This article tackles a subject that has not been previously explored in depth and proposes a structured approach to addressing health risks in professional esports. The insights provided contribute to an important discussion in the field.
• The structure of the article could be improved, as the manuscript lacks a dedicated Discussion section (despite it being described in PRISMA list as lines 157-339), with the Results section seemingly fulfilling both roles. In a literature review, it is generally expected that results are presented separately from their interpretation and broader implications. The current format makes it difficult to distinguish between the summary of findings from the reviewed studies and the authors' critical analysis.
• The manuscript would benefit from a brief discussion of its limitations, particularly regarding potential biases in literature selection, the scope of included studies, or gaps in existing research. Acknowledging these aspects would provide a more balanced perspective and help contextualize the findings within the broader field of esports health management.

---

## Round 0.2 · accepted · Accept

Dear Authors
Thank you for making all the suggested changes
Congratulations!

·

Basic reporting

No new comments from previous review.

Experimental design

With the inclusion of additional sources, as well as addressing the concerns of Reviewers 2 and 3, I find sufficient improvement to recommend publication.

Validity of the findings

No new comments from previous review.

·

Basic reporting

The revised manuscript reflects a commendable effort by the authors to address the initial concerns. I appreciate the careful consideration given to the feedback and the substantial improvements evident throughout the text. Overall, the revisions contribute significantly to the article’s impact and readability, marking a clear advancement from the earlier draft.

Experimental design

no comment

Validity of the findings

no comment